# Prevalence of dermatological manifestations due to face mask use and its associated factors during COVID-19 among the general population of Bangladesh: A nationwide cross-sectional survey

Simanta Roy[1,2], Mohammad Azmain Iktidar[1,2], Sreshtha Chowdhury[1,2], A. M. Khairul Islam[1,2], Auditia Deb[3], Shresta Chowdhury[4], Shahidur Rahman[3], Madhuritu Bhadra Medha[5], Antara Das Gupta[6], Afia Tasnim[3], Rifat Ara[1,7]*, Mohammad Delwer Hossain Hawlader[1]

1 Department of Public Health, North South University, Dhaka, Bangladesh, 2 Public Health Professional Development Society (PPDS), Dhaka, Bangladesh, 3 Chittagong Medical College and Hospital, Chattogram, Bangladesh, 4 Dhaka Medical College, Dhaka, Bangladesh, 5 ZH Sikder Women's Medical College & Hospital, Dhaka, Bangladesh, 6 Rangamati Medical College, Chattogram, Bangladesh, 7 Infectious Disease Division, icddr,b, Dhaka, Bangladesh

* rifat.ara@icddrb.org

## Abstract

Following the worldwide COVID-19 pandemic, individuals have begun to take preventive measures to avoid exposure. Among the precautionary measures, facemask was mostly emphasized. This study aimed to determine the prevalence of dermatological symptoms linked with face mask usage and explore other associated factors. This cross-sectional survey was conducted throughout all eight divisions of Bangladesh. 1297 people were approached using a fixed-step procedure on a random route sample where 803 fulfilled the inclusion criteria. The overall prevalence of dermatological manifestation in this study was 40.85%. The common dermatological manifestations due to facemasks use were acne (26%), allergy symptoms (24%), traumatic symptoms (24%), and other symptoms (26%). Two important frequently reported risk factors were previous history of skin diseases and obesity. Females were more likely to have acne (CI: 1.199, 3.098; p = .007) and allergy issues (CI: 1.042, 2.359; p = .031). N95 and KN95 masks were more likely to produce allergic symptoms, while surgical mask users were more likely to develop acne. Acne was prevalent more than twice (CI: 1.42, 4.26; p = 0.001) in persons with a COVID-19 infection history. Further exploration is required to find out the reason. Surgical mask users reported more complaints than other types of masks, and prolonged use caused more skin symptoms. Modifications in the pattern of facemask usage and planning for work recesses might also be advised to provide for a pause from uninterrupted facemask use.

**Data Availability Statement:** All relevant data are within the paper and its Supporting Information files.

**Funding:** The author(s) received no specific funding for this work.

**Competing interests:** The authors have declared that no competing interests exist.

## Introduction

The tradition of covering the mouth and nose dates back to early modern Europe. In order to counteract the airborne miasma, physicians used to wear bird-like plague doctor masks filled with perfumes and spices. This technique faded by the 18th century, and modern face masks were invented after that [1]. In 1897, a study was conducted on protective face masks in operating rooms by Carl Friedrich Flügge and Johannes von Mikulicz, where a "mouth bandage" of gauze was used to protect the patients from wound infection [2, 3]. Face masks are currently used not just during performing surgeries but also to prevent the spread of respiratory infections like the Swine flu (2009) and SARS (2003) [4, 5].

WHO issued a public health emergency on January 30, 2020, due to novel coronavirus (SARS-CoV-2). On March 11, 2020, the illness was declared a pandemic and termed COVID-19 [6, 7]. The most likely method of transmission seems to be droplets created by face-to-face contact while chatting, coughing, or sneezing [8]. The results of over 30 researches across the globe were evaluated in detail and revealed a statistically significant decrease in the incidence of COVID-19 to 62% with mask usage and a 31% reduction by maintaining physical distancing [9]. Vaccines are safe and effective, and they are saving lives worldwide. However, the majority do not provide 100% protection, the majority of nations have not vaccinated everyone, and it is unknown if vaccines will prevent future transmission of new coronavirus strains. Hence, facemasks have become the most popular and efficient tools in preventing COVID-19 outbreaks.

Acne, rash, itching, xerosis, and nasal bridge scarring are some dermatological manifestations that have been documented among face mask users [10–12]. Long-term usage of masks, the impacts of various fabric materials, and the varieties in surgical mask quality have made dermatological problems more complicated in recent days. Interestingly, N95 and surgical facemasks may produce variable temperatures and humidity in the microclimates of this protective equipment, which can significantly impact heart rate, thermal stress, and subjective impression of pain [13]. According to some other researches, exposure time seems to be the most significant risk factor for facial dermatitis, especially when wearing masks for more than 6 hours [14]. The dermatological reactions observed by the medical professionals and mass people include pressure dermatitis due to the use of masks and helmets, contact irritants or allergic dermatitis of the hands, excessive sweating, bacterial infections, and acne due to the use of face masks [15]. Pre-existing skin conditions such as xerosis cutis, seborrheic dermatitis, acne, and urticaria may worsen by using face coverings [16].

Since the pandemic, healthcare providers, as well as the general population all around the world, started using facial masks regularly, and it has become a new normal now. Bangladesh is also not an exception in case of following the guidance of protective measures given by the World Health Organization. As a result of this utilization of facemask for a long time, complaints of abnormal dermatological presentations exist. Given that significant aspect of facemask-related cutaneous effects that are unidentified, this research looked at the prevalence and impacts of different types of facemask-related dermatological symptoms in the general population of Bangladesh. This study also looked at some probable determinants of such issues.

## Methods

### Study design, site, and instrument

The research used a cross-sectional method of survey among the general population in eight divisions of Bangladesh: Dhaka, Chattogram, Rajshahi, Khulna, Sylhet, Barisal, Mymensingh, and Rangpur. Data were collected between May and July 2021. Facemask-related dermatological symptoms were assessed using a structured questionnaire that included information about

the respondent's sociodemographic situation (age, sex, occupation, and workplace), the duration of mask use (<12 months or >12 months), and the average weekly duration mask use (32 hours or 32–56 hours or >56 hours). History of co-morbidities (e.g., diabetes, hypertension, asthma, obesity etc.) were self-reported and only those were included in the study who were confirmed by any healthcare professional in the past. Additionally, the type of facemask they prefer (N95, KN95, surgical, cloth, or other masks (e.g., snorkel face mask, sponge facemask, full face mask), their concurrent use of multiple masks, cooling and ventilation system of their workplaces, and the history of COVID-19 infection and vaccinations were included in the questionnaire. Moreover, the number of muggier months (8 months, 8–9 months, >9 months) of the year was established based on their residence [17]. The dew point was used to assess the humidity comfort level since it influences whether sweat evaporates off the skin, cooling the body. A pilot test was carried out prior to the main study to validate the questionnaire.

## Participants and sampling

A fixed-step procedure (every fifth person) on a random route sampling method was used in this study. Quota sampling technique based on gender was also used to create a representative sample of the general population by city (using data from the Bangladesh Bureau of Statistics). For instance, if a male subject is required to meet the quota, every fifth individual is approached until a male is located. To create divisionally representative data, 100 participants from each division were interviewed. Total 1,297 people at renowned public places of eight cities were approached, and 803 participants were included in this study who met the eligibility criteria. Nonmedical volunteers were trained as interviewers and were required to adhere to a predefined neutral script to avoid selection bias. Consenting of all included participants were done before collecting data from them.

For eligibility assessment, adult (above 18 years) individuals who wear facemasks (irrespective of types) in public places and in their workplace on a regular basis were included in this study since the primary objective of this research was to analyze the dermatological symptoms associated with prolong and regular use of facemask. Participants were excluded who were foreign nationals residing in Bangladesh, working as healthcare workers, or expressing unwillingness to participate in this study.

## Statistical analysis

The data were analyzed using STATA software version 16. Quantitative data were summarized using mean, a measure of center, and standard deviation as a measure of variability. Categorical variables are expressed as frequency with relative frequency. The chi-square test was performed to investigate the bivariate relationship between categorical variables, and logistic regression models were fitted to identify factors related to outcomes.

## Ethical consideration

The study protocol was authorized by North South University's Institutional Review Board (IRB no: 2021/OR-NSU/IRB/1001) and adhered to the 1975 Declaration of Helsinki's ethical criteria (6th version, 2008), as shown in a priori approval by the institutional review committee.

## Result

803 participants out of 1,297 responded to the full questionnaire, with almost equal participation of male and female. Highest participation was noticed from the young adult population

group. All the significant descriptive characteristics of the study participants have been summarized in Table 1.

Several qualities of masks have been used by the participants where surgical masks were employed mostly (688; 85.68%) after that cotton masks were more preferable (309; 38.48%) than N95 masks (34; 4.23%) and KN95 (119; 14.82%). 87.05% of the mask users were using facemasks for 12 months or more, while 12.95% were using them for less than 12 months. In case of weekly duration of facemasks use, 24.28% of participants use them > 32 hours per week, 30.14% for 8 to 32 hours, and 45.58% for < 8 hours per week. Significantly, 39.23% of the participants use double masks, and 50% of respondents never repeat the mask after first-time use. Additionally, 14.69% of participants had COVID-19 infection, whereas 50.06% were fully vaccinated.

Significant dermatological manifestations due to the use of facemasks that have been accounted from the participating individuals include acne (26%), allergic symptoms (24%), traumatic symptoms (24%), and 26% were other symptoms (Fig 1). Among the allergic symptom, rashes, itching, and redness were major complaints, whereas cracked skin, blistering skin, and pressure sore were the notable traumatic complaints. Dryness of the skin and skin color changes were some other symptoms that were recorded from the participants.

The unadjusted findings of the bivariate analysis are reported in Table 2. The results show that the possible factors gender, duration of facemask usage, KN95, and N95 types of facemasks were significantly associated with allergic manifestations. According to the analysis, potential variables such as employment status, working hours per week, humid months in the previous 12 months, average facemask use per week, cloth facemask, and facemask reuse pattern were significantly associated with traumatic manifestations. Age, gender, education level, employment status, monthly family income, working hours per week, average facemask usage per week, surgical and N95 types of masks, and COVID-19 infection were significantly associated with acne breakouts. In addition, the variables average facemask usage per week, cloth facemask, and facemask reuse pattern were significantly associated with other skin manifestations.

## Allergic manifestations

In our multivariate logistic regression model, we included all the potential variables that were established in bivariate analysis. With this analysis, we incorporated the adjusted result and showed it in Table 3. Women had a 56% higher incidence of allergy symptoms than men [95% CI: 1.042, 2.359, and p = 0.031]. Those already suffering from skin illnesses were 86% more likely to have allergies than healthy individuals [95% CI: 1.109, 3.125 and p = 0.019]. Obese individuals had 79% more tendencies to develop allergic symptoms than non-obese people [95% CI: 1.035, 3.095 and p = 0.037]. The KN95 mask users showed 67% higher risk [95% CI: 1.01, 2.772 and p = 0.046], and N95 mask users are 2.62 times more likely [95% CI: 1.22, 5.809, and p = 0.014] to develop allergies than other types of mask users. Complaints of allergies were 62% less likely in those who used facemasks for over a year [95% CI: .233, .642 and p<0.001]. Those who wear masks for 8–32 hours per week were 80% more likely to suffer from allergic manifestations than those who use less than 8 hours per week [95% CI: 1.47–2.82 and p = 0.110].

## Traumatic manifestations

Traumatic difficulties were 3.44 times more likely to occur in participants with a prior skin disease history (95% CI: 2.138, 5.54, p<0.001). Obese individuals had 2.28 times the risk of traumatic symptoms than non-obese (95% CI: 1.3583, 3.827 and p = 0.002). Working with masks

**Table 1. Descriptive characteristics of the study participants (N = 803).**

| Variable | N | % |
|---|---|---|
| **Age** | | |
| ≤ 20 years | 211 | 26.28 |
| 21–35 years | 369 | 45.95 |
| > 35 years | 223 | 27.77 |
| **Gender** | | |
| Male | 409 | 50.93 |
| Female | 394 | 49.07 |
| **Education** | | |
| Uneducated | 20 | 2.49 |
| Higher secondary | 303 | 37.73 |
| Graduation | 370 | 46.08 |
| Post-graduation | 110 | 13.70 |
| **Employment status** | | |
| Unemployed | 542 | 67.50 |
| Employed | 261 | 32.50 |
| **Family income per month (in bdt)** | | |
| <30000 | 202 | 25.16 |
| 30000–60000 | 383 | 47.70 |
| >60000 | 218 | 27.15 |
| **Working hours** | | |
| Not applicable | 274 | 34.12 |
| < 24 hours/week | 293 | 36.49 |
| 24–48 hours/week | 107 | 13.33 |
| > 48 hours/week | 129 | 16.06 |
| **Workplace Air Conditioning** | | |
| No | 639 | 79.58 |
| Yes | 164 | 20.42 |
| **Religion** | | |
| Islam | 543 | 67.62 |
| Hinduism | 232 | 28.89 |
| Buddhism | 28 | 3.49 |
| **Marital status** | | |
| Unmarried | 499 | 62.14 |
| Married | 304 | 37.86 |
| **Humid months in last 12 months** | | |
| < 8 months | 21 | 2.62 |
| 8–9 moths | 168 | 20.92 |
| > 9 months | 614 | 76.46 |
| **Comorbidities/Risk factors** | | |
| **Diabetes** | | |
| Yes | 85 | 10.59 |
| No | 718 | 89.41 |
| **Skin disease** | | |
| Yes | 96 | 11.96 |
| No | 707 | 88.04 |
| **Obesity** | | |
| Yes | 84 | 10.46 |
| No | 719 | 89.54 |

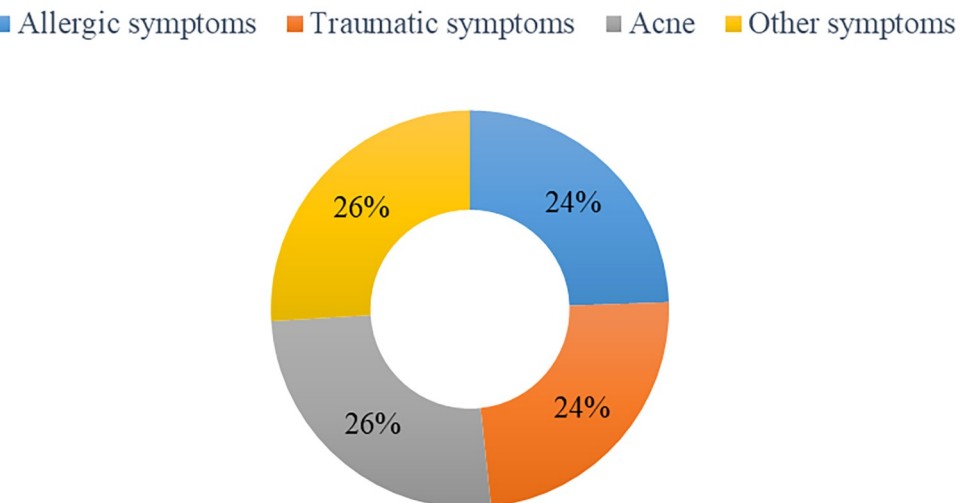

**Fig 1. Prevalence of different dermatological symptoms due to facemask use.**

for 8–32 hours per week increased traumatic symptoms by 76% [95% CI: 1.12, 2.75 and p = 0.013]. Participants who wore masks for more than 32 hours per week were 2.03 times more likely to have traumatic manifestations than those who worked less than 8 hours per week [95% CI: 1.24, 3.306 and p = 0.004].

## Acne manifestations

Females had a 92% higher risk of acne than males [95% CI: 1.199, 3.098, p = 0.007]. Participants with a previous history of skin diseases had a 79% greater likelihood of getting acne than healthy participants [95% CI: 1.017, 3.159 and p = 0.044]. Obese people had 2.16 times more acne breakouts than non-obese people [95% CI: 1.44, 4.111, p = 0.018]. The surgical mask users showed a 2.40 times greater risk of acne than non-users [95% CI: 1.076, 5.38, and p = 0.032]. The N95 mask users had three times more acne problems than non-users [95% CI: 1.224, 7.388, and p = 0.016]. Previous COVID infection increased acne risk by 2.29 [95% CI: 1.35, 3.902, and p = 0.002].

Participants who had a previous history of skin disease, having other skin problems were 2.48 times higher among them [95% CI: 1.42, 4.26 and p = 0.001]. Participants who used the cloth mask had 59% more other skin problems than those who used other types of facemasks [95% CI: 1.005, 2.526, and p = 0.048].

## Discussion

Bangladesh is a tropical country with warm and humid climates. The risk of cellulitis, contact dermatitis, and heat rashes are already more prone in sweltering weather [18]. Wearing loose, breathable clothing and maintaining a cool environment are usually suggested by dermatologists to prevent abnormal skin manifestations. The use of facemasks was always applicable based on different purposes, but the regular use of it has been initiated for the last one and half years. Wearing a mask is essential for frontline healthcare workers since it allows them to fight the deadly COVID-19 with less worry of getting the infection. For the common citizen, however, the use of masks has currently become the most effective psychological emblem [19]. It is

**Table 2. Facemask use related skin manifestations and associated factors (N = 803).**

| Variable | Allergic manifestations | | | Traumatic manifestations | | | Acne | | | Others | | |
|---|---|---|---|---|---|---|---|---|---|---|---|---|
| | No | Yes | P value | No | Yes | P value | No | Yes | P value | No | Yes | P value |
| **Age** | | | | | | | | | | | | |
| ≤ 20 years | 169(80.09) | 42(19.1) | 0.455 | 176(83.41) | 35(16.59) | 0.235 | 162(76.78) | 49(23.22) | **<0.001** | 187(88.63) | 24(11.37) | 0.568 |
| 21–35 years | 298(80.76) | 71(19.24) | | 286(77.51) | 83(22.49) | | 312(84.55) | 57(15.45) | | 319(86.45) | 50(13.55) | |
| > 35 years | 188(84.30) | 35(15.70) | | 178(79.82) | 45(20.18) | | 216(96.86) | 7(3.14) | | 190(85.20) | 33(14.80) | |
| **Gender** | | | | | | | | | | | | |
| Male | 345(84.35) | 64(15.65) | **0.038** | 320(78.24) | 89(21.76) | 0.294 | 372(90.95) | 37(9.05) | **<0.001** | 357(87.29) | 52(12.71) | 0.604 |
| Female | 310(78.68) | 84(21.32) | | 320(81.22) | 74(18.78) | | 318(80.71) | 76(19.29) | | 339(86.04) | 55(13.96) | |
| **Education** | | | | | | | | | | | | |
| Uneducated | 18(90.00) | 2(10.00) | 0.610 | 17(85.00) | 3(15.00) | 0.828 | 20(100.00) | 0(0.00) | **0.003** | 16(80.00) | 4(20.00) | 0.319 |
| Higher secondary | 243(80.20) | 60(19.80) | | 244(80.53) | 59(19.47) | | 270(89.11) | 33(10.89) | | 256(84.49) | 47(15.51) | |
| Graduation | 306(82.70) | 64(17.30) | | 294(79.46) | 76(20.54) | | 301(81.35) | 69(18.65) | | 325(87.84) | 45(12.16) | |
| Post-graduation | 88(80.00) | 22()20.00 | | 85(77.27) | 25(22.73) | | 99(90.00) | 11(10.00) | | 99(90.00) | 11(10.00) | |
| **Employment status** | | | | | | | | | | | | |
| Unemployed | 443(81.73) | 99(18.27) | 0.860 | 445(82.10) | 97(17.90) | **0.015** | 443(81.73) | 99(18.27) | **<0.001** | 476(87.82) | 66(12.18) | 0.168 |
| Employed | 212(81.23) | 49(18.77) | | 195(74.71) | 66(25.29) | | 247(94.64) | 14(5.36) | | 220(84.29) | 41(15.71) | |
| **Family income per month(in bdt)** | | | | | | | | | | | | |
| <30000 | 165(81.68) | 37(18.32) | 0.880 | 158(78.22) | 44(21.78) | 0.666 | 186(92.08) | 16(7.92) | **0.013** | 170(84.16) | 32(15.84) | 0.144 |
| 30000–60000 | 310(80.94) | 73(19.06) | | 304(79.37) | 79(20.63) | | 319(83.29) | 64(16.71) | | 329(85.90) | 54(14.10) | |
| >60000 | 180(82.57) | 38(17.43) | | 178(81.65) | 40(18.35) | | 185(84.86) | 33(15.14) | | 197(90.37) | 21(9.63) | |
| **Working hours per week** | | | | | | | | | | | | |
| Not applicable | 231(84.31) | 43(15.69) | 0.090 | 232(84.67) | 42(15.33) | **0.011** | 225(82.12) | 49(17.88) | **0.006** | 239(87.23) | 35(12.77) | 0.065 |
| < 24 hours | 234(79.86) | 59(20.14) | | 236(80.55) | 57(19.45) | | 248(84.64) | 45(15.36) | | 262(89.42) | 31(10.58) | |
| 24–48 hours | 80(74.77) | 27(25.23) | | 78(72.90) | 29(27.10) | | 95(88.79) | 12(11.21) | | 92(85.98) | 15(14.02) | |
| > 48 hours | 110(85.27) | 19(14.73) | | 94(72.87) | 35(27.13) | | 122(94.57) | 7(5.43) | | 103(79.84) | 26(20.16) | |
| **Humid months in last 12 months** | | | | | | | | | | | | |
| < 8 months | 18(85.71) | 3(14.29) | 0.720 | 18(85.71) | 3(14.29) | **0.019** | 18(85.71) | 3(14.29) | 0.830 | 18(85.71) | 3(14.29) | 0.639 |
| 8–9 moths | 134(79.76) | 34(20.24) | | 121(72.02) | 47(27.98) | | 142(84.52) | 26(15.48) | | 142(84.52) | 26(15.48) | |
| > 9 months | 503(81.92) | 111(18.08) | | 501(81.60) | 113(18.40) | | 530(86.32) | 84(13.68) | | 536(87.30) | 78(12.70) | |
| **Duration of facemask use** | | | | | | | | | | | | |
| ≤ 12 months | 74(71.15) | 30(28.85) | **0.003** | 80(76.92) | 24(23.08) | 0.450 | 90(86.54) | 14(13.46) | 0.840 | 86(82.69) | 18(17.31) | 0.200 |
| > 12 months | 581(83.12) | 118(16.88) | | 560(80.11) | 139(19.89) | | 600(85.84) | 99(14.16) | | 610(87.27) | 89(12.73) | |
| **Average facemask use per week** | | | | | | | | | | | | |
| < 8 hours | 308(84.15) | 58(15.85) | 0.130 | 312(85.25) | 54(14.75) | **0.001** | 298(81.42) | 68(18.58) | **<0.001** | 324(88.52) | 42(11.48) | **0.029** |
| 8–32 hours | 188(77.69) | 54(22.31) | | 185(76.45) | 57(23.55) | | 210(86.78) | 32(13.22) | | 214(88.43) | 28(11.57) | |
| > 32 hours | 159(81.54) | 36(18.46) | | 143(73.33) | 52(26.67) | | 182(93.33) | 13(6.67) | | 158(81.03) | 37(18.97) | |
| **Surgical facemask** | | | | | | | | | | | | |
| No | 95(82.61) | 20(17.39) | 0.750 | 94(81.74) | 21(18.26) | 0.557 | 107(93.04) | 8(6.96) | **0.018** | 99(86.09) | 16(13.91) | 0.841 |
| Yes | 560(81.40) | 128(18.60) | | 546(79.36) | 142(20.64) | | 583(84.74) | 105(15.26) | | 597(86.77) | 91(13.23) | |
| **Cloth facemask** | | | | | | | | | | | | |
| No | 410(83.00) | 84(17.00) | 0.180 | 408(82.59) | 86(17.41) | **0.010** | 428(86.64) | 66(13.36) | 0.460 | 442(89.47) | 52(10.53) | **0.003** |
| Yes | 245(79.29) | 64(20.71) | | 232(75.08) | 77(24.92) | | 262(84.79) | 47(15.21) | | 254(82.20) | 55(17.80) | |
| **KM95** | | | | | | | | | | | | |
| No | 569(83.19) | 115(16.81) | **0.005** | 548(80.12) | 136(19.88) | 0.482 | 589(86.11) | 95(13.89) | 0.720 | 596(87.13) | 88(12.87) | 0.350 |
| Yes | 86(72.27) | 33(27.73) | | 92(77.31) | 27(22.69) | | 101(84.87) | 18(15.13) | | 100(84.03) | 19(15.97) | |
| **N95** | | | | | | | | | | | | |

*(Continued)*

**Table 2.** (Continued)

| Variable | Allergic manifestations | | | Traumatic manifestations | | | Acne | | | Others | | |
|---|---|---|---|---|---|---|---|---|---|---|---|---|
| | No | Yes | P value | No | Yes | P value | No | Yes | P value | No | Yes | P value |
| No | 635(82.57) | 134(17.43) | <0.001 | 614(79.84) | 155(20.16) | 0.632 | 666(86.61) | 103(13.39) | **0.008** | 670(87.13) | 99(12.87) | 0.070 |
| Yes | 20(58.82) | 14(41.18) | | 26(76.47) | 8(23.53) | | 24(70.59) | 10(29.41) | | 26(76.47) | 8(23.53) | |
| **Facemask reuse pattern** | | | | | | | | | | | | |
| Reuse without cleaning | 38(86.36) | 6(13.64) | 0.280 | 37(84.09) | 7(15.91) | **0.023** | 40(90.91) | 4(9.09) | 0.610 | 41(93.18) | 3(6.82) | **0.042** |
| Reuse after cleaning | 283(79.27) | 74(20.73) | | 269(75.35) | 88(24.65) | | 306(85.71) | 51(14.29) | | 298(83.47) | 59(16.53) | |
| Single use | 334(83.08) | 68(16.92) | | 334(83.08) | 68(16.92) | | 344(85.57) | 58(14.43) | | 357(88.81) | 45(11.19) | |
| **Simultaneous multiple facemask use** | | | | | | | | | | | | |
| No | 404(82.79) | 84(17.21) | 0.260 | 395(80.94) | 93(19.06) | 0.276 | 424(86.89) | 64(13.11) | 0.330 | 428(87.70) | 60(12.30) | 0.280 |
| Yes | 251(79.68) | 64(20.32) | | 245(77.78) | 70(22.22) | | 266(84.44) | 49(15.56) | | 268(85.08) | 47(14.92) | |
| **COVID-19 infection** | | | | | | | | | | | | |
| No | 564(82.34) | 121(17.66) | 0.170 | 552(80.58) | 133(19.42) | 0.134 | 600(87.59) | 85(12.41) | **0.001** | 598(87.30) | 87(12.70) | 0.200 |
| Yes | 91(77.12) | 27(22.88) | | 88(74.58) | 30(25.42) | | 90(76.27) | 28(23.73) | | 98(83.05) | 20(16.95) | |
| **COVID-19 vaccination status** | | | | | | | | | | | | |
| Not started | 250(81.97) | 55(18.03) | 0.320 | 243(79.67) | 62(20.33) | 0.775 | 266(87.21) | 39(12.79) | 0.270 | 261(85.57) | 44(14.43) | 0.440 |
| Only 1st dose | 73(76.04) | 23(23.96) | | 74(77.08) | 22(22.92) | | 86(89.58) | 10(10.42) | | 87(90.63) | 9(9.38) | |
| Both doses | 332(82.59) | 70(17.41) | | 323(80.35) | 79(19.65) | | 338(84.08) | 64(15.92) | | 348(86.57) | 54(13.43) | |

All data presented as N (%), Pearson Chi square test was done, p-values <0.05 are significant

also said that mask use negates all other infection prevention recommendations for interrupting the COVID-19 transmission chain, such as hand washing, personal cleanliness, and social distancing [19]. A novel perspective approach to explore the pros and cons of facemask use has been performed in various community settings of different countries, which revealed several opinions and preferences regarding mask use [20]. Suggestions to search for alternatives of facemask were also raised in the study for the patients with COPD, acute and chronic respiratory disease, older age, underlying medical conditions, and hypercapnia sensitive group.

Skin damages due to prolong facemask use have become a universal hurdle now. Contact dermatitis, pressure erythema, even eczematous lesions are some severe forms of dermatological problems that have been reported due to protective equipment uses [14]. In our study, four categories of dermatological complaints were significant among the general population, and manifestation rates were closer. To see the influence of tropical weather, enumeration of muggier months has been done, where three-fourth of the participants stated living in humid areas for a long time. "Maskne", which is the new term of acne that occurs due to facial masks or coverings [21], was also found as the prime dermatological complaint among general inhabitants who participated in our study. Here, gender, obesity, and preceding skin diseases have been found as some of the important aggravating factors of acne. Most of the females, obese participants, as well as individuals who are already suffering from skin problems, came up with a history of the flare of acne. Moreover, complaints of acne were two to three times higher among the surgical and N95 facemask users. Interestingly, individuals who once got infected with COVID-19, reported more about the acne breakout. Further exploration regarding this finding is necessary.

Allergic and traumatic manifestations are two other important dermatological problems that have been reported by our study participants. In these two cases, obesity was a significant variable to provoke the symptoms. It is already found that obese people are more prone to develop allergic reactions, as they usually induce a reduction in immune tolerance [22].

**Table 3. Multivariate logistic regression analysis of face mask related skin manifestations and associated factors (N = 803).**

| | Allergic symptoms | | | | Traumatic symptoms | | | | Acne | | | | Other symptoms | | | |
|---|---|---|---|---|---|---|---|---|---|---|---|---|---|---|---|---|
| | AOR | p-value | 95% CI | | AOR | p-value | 95% CI | | AOR | p-value | 95% CI | | AOR | p-value | 95% CI | |
| **Gender** | | | | | | | | | | | | | | | | |
| Male | 1 | | | | 1 | | | | 1 | | | | 1 | | | |
| Female | 1.568 | **0.031** | 1.042 | 2.359 | 0.893 | 0.578 | 0.6 | 1.329 | 1.927 | **0.007** | 1.199 | 3.098 | 1.185 | 0.469 | 0.748 | 1.879 |
| **Comorbidity** | | | | | | | | | | | | | | | | |
| **Diabetes** | | | | | | | | | | | | | | | | |
| No | 1 | | | | | | | | 1 | | | | | | | |
| Yes | 1.223 | 0.556 | 0.625 | 2.395 | | | | | 0.689 | 0.584 | 0.182 | 2.612 | | | | |
| **Skin disease** | | | | | | | | | | | | | | | | |
| No | 1 | | | | 1 | | | | 1 | | | | 1 | | | |
| Yes | 1.861 | **0.019** | 1.109 | 3.125 | 3.443 | **<0.001** | 2.138 | 5.544 | 1.792 | **0.044** | 1.017 | 3.159 | 2.488 | **0.001** | 1.452 | 4.265 |
| **Obesity** | | | | | | | | | | | | | | | | |
| No | 1 | | | | 1 | | | | 1 | | | | 1 | | | |
| Yes | 1.79 | **0.037** | 1.035 | 3.095 | 2.28 | **0.002** | 1.358 | 3.827 | 2.168 | **0.018** | 1.144 | 4.111 | 1.649 | 0.110 | 0.893 | 3.047 |
| **Facemask type** | | | | | | | | | | | | | | | | |
| **Cloth facemask** | | | | | | | | | | | | | | | | |
| No | 1 | | | | | | | | 1 | | | | 1 | | | |
| Yes | 1.147 | 0.517 | 0.757 | 1.738 | | | | | 1.219 | 0.412 | 0.76 | 1.955 | 1.593 | **0.048** | 1.005 | 2.526 |
| **Surgical mask** | | | | | | | | | | | | | | | | |
| No | 1 | | | | 1 | | | | 1 | | | | 1 | | | |
| Yes | 1.086 | 0.781 | 0.608 | 1.94 | 1.155 | 0.616 | 0.658 | 2.026 | 2.407 | **0.032** | 1.076 | 5.384 | 1.045 | 0.891 | 0.556 | 1.964 |
| **KN95** | | | | | | | | | | | | | | | | |
| No | 1 | | | | 1 | | | | 1 | | | | 1 | | | |
| Yes | 1.673 | **0.046** | 1.01 | 2.772 | 1.121 | 0.668 | 0.666 | 1.887 | 0.685 | 0.232 | 0.369 | 1.273 | 1.36 | 0.297 | 0.763 | 2.423 |
| **N95** | | | | | | | | | | | | | | | | |
| No | 1 | | | | 1 | | | | 1 | | | | | | | |
| Yes | 2.662 | **0.014** | 1.22 | 5.809 | 1.06 | 0.898 | 0.434 | 2.591 | 3.007 | **0.016** | 1.224 | 7.388 | | | | |
| **Duration of use** | | | | | | | | | | | | | | | | |
| ≤ 12 months | 1 | | | | 1 | | | | | | | | 1 | | | |
| > 12 months | 0.387 | **<0.001** | .233 | 0.642 | 0.651 | 0.116 | 0.381 | 1.111 | | | | | 0.646 | 0.143 | 0.36 | 1.159 |
| **Use per week** | | | | | | | | | | | | | | | | |
| < 8 hours | 1 | | | | 1 | | | | 1 | | | | 1 | | | |
| 8–32 hours | 1.801 | **0.011** | 1.147 | 2.828 | 1.763 | **0.013** | 1.127 | 2.757 | 0.905 | 0.699 | 0.547 | 1.497 | 1.022 | 0.936 | 0.597 | 1.75 |
| > 32 hours | 1.673 | 0.071 | 0.957 | 2.925 | 2.031 | **0.004** | 1.247 | 3.306 | 0.696 | 0.321 | 0.341 | 1.423 | 1.877 | **0.023** | 1.093 | 3.224 |
| **Use type** | | | | | | | | | | | | | | | | |
| Single-use | 1 | | | | 1 | | | | 1 | | | | 1 | | | |
| Reuse without cleaning | 0.706 | 0.463 | 0.279 | 1.788 | 0.837 | 0.691 | 0.347 | 2.016 | 0.561 | 0.306 | 0.185 | 1.698 | 0.526 | 0.307 | 0.154 | 1.804 |
| Reuse after cleaning | 1.143 | 0.538 | 0.747 | 1.747 | 1.665 | **0.011** | 1.126 | 2.462 | 1.063 | 0.802 | 0.659 | 1.716 | 1.284 | 0.300 | 0.8 | 2.06 |
| **Multiple masks** | | | | | | | | | | | | | | | | |
| No | 1 | | | | 1 | | | | | | | | 1 | | | |
| Yes | 1.115 | 0.583 | 0.756 | 1.644 | 1.25 | 0.245 | 0.858 | 1.821 | | | | | 1.275 | 0.270 | 0.828 | 1.965 |
| **History of COVID** | | | | | | | | | | | | | | | | |
| No | 1 | | | | 1 | | | | 1 | | | | 1 | | | |
| Yes | 1.316 | 0.282 | 0.798 | 2.17 | 1.43 | 0.148 | 0.881 | 2.322 | 2.296 | **0.002** | 1.351 | 3.902 | 1.351 | 0.287 | 0.777 | 2.348 |
| **COVID Vaccination status** | | | | | | | | | | | | | | | | |

(*Continued*)

**Table 3.** (Continued)

| | Allergic symptoms | | | Traumatic symptoms | | | Acne | | | Other symptoms | | |
|---|---|---|---|---|---|---|---|---|---|---|---|---|
| | AOR | p-value | 95% CI | | AOR | p-value | 95% CI | | AOR | p-value | 95% CI | | AOR | p-value | 95% CI | |
| Not vaccinated | 1 | | | | 1 | | | | | | | | | | | |
| Only 1st dose | 1.416 | 0.251 | 0.782 | 2.566 | 1.222 | 0.507 | 0.675 | 2.214 | | | | | | | | |
| Both doses | 0.981 | 0.932 | 0.633 | 1.519 | 1.059 | 0.787 | 0.701 | 1.598 | | | | | | | | |

AOR = Adjusted Odds Ratio, CI = Confidence Interval

p-values <0.05 are significant

Additionally, those who used masks more than 8 hours per week tended to develop more allergies and scars in our study. As the study says, that prolonged use of facemask in a hot environment can aggravate dyspnea [23], so the prohibition of long-term facemask use or interval can be recommended to see whether this problem can be minimized or not.

From the opinions about the preference of face mask types, cotton cloth masks were more preferable and comfortable among our study participants. Rather, dermatological problems were profound among the surgical facemask users. This outcome is noteworthy, as this predilection significantly can help in decreasing the demand for surgical masks among general people and should be reserved for the healthcare providers during the COVID-19 pandemic. A prospective survey was conducted in Thailand, where the effects of facemask use have been compared among the healthcare providers and non-healthcare general citizens [24]. This survey showed some similar outcomes along with a higher risk of skin problems among the healthcare workers. This is obvious, as health care workers (HCW) use facemask more frequently and for a longer duration than the non-HCWs. However, several factors are also responsible for the general population to suffer from skin irritations due to face mask use, such as the types and extent of their work.

As this study was conducted all over the country, recruiting participants from all eight divisions, it represents a nationwide result. This was the strength regarding the outcome of our study. Our limitation was collecting information about dermatological symptoms based only on the participants' statements; assessment by dermatology experts could not be done in our study.

## Conclusion

The overall prevalence of dermatological manifestations due to the use of facemask was found 40.85% among the general people in our study. As the weather and humidity in all divisions of Bangladesh are almost similar, no significant difference was found associated with the duration of the muggier months. Surgical mask users had more objections rather than the other types of facemask users, and longer usage duration created more skin manifestations, which is very obvious. As previous skin diseases and obesity came out two important covariates in our study, dermatological experts should investigate further to sort out the solutions for these groups of people. Moreover, changes in the pattern of facemask use and planning for recesses in the workplace can be recommended, which can create a minimum comfort to take a break from continuous facemask wearing.

## Supporting information

**S1 File. Complete data set of this study.** Complete data set of this study.
(PDF)

## Acknowledgments

Authors would like to acknowledge the technical supports that was provided by the Research Aid Bangladesh (RAB) during conducting the study.

## Author Contributions

**Conceptualization:** Simanta Roy, Mohammad Azmain Iktidar, Sreshtha Chowdhury, A. M. Khairul Islam, Rifat Ara, Mohammad Delwer Hossain Hawlader.

**Data curation:** Simanta Roy, Mohammad Azmain Iktidar, Sreshtha Chowdhury, A. M. Khairul Islam, Auditia Deb, Shresta Chowdhury, Shahidur Rahman, Madhuritu Bhadra Medha, Antara Das Gupta, Afia Tasnim.

**Formal analysis:** Simanta Roy, Mohammad Azmain Iktidar, Sreshtha Chowdhury, A. M. Khairul Islam, Auditia Deb, Shresta Chowdhury, Shahidur Rahman, Madhuritu Bhadra Medha, Antara Das Gupta, Afia Tasnim, Rifat Ara.

**Investigation:** Simanta Roy, Mohammad Azmain Iktidar, Auditia Deb, Shresta Chowdhury, Shahidur Rahman, Madhuritu Bhadra Medha, Antara Das Gupta, Afia Tasnim, Rifat Ara, Mohammad Delwer Hossain Hawlader.

**Methodology:** Simanta Roy, Mohammad Azmain Iktidar, Sreshtha Chowdhury, A. M. Khairul Islam, Auditia Deb, Shresta Chowdhury, Shahidur Rahman, Madhuritu Bhadra Medha, Antara Das Gupta, Afia Tasnim, Mohammad Delwer Hossain Hawlader.

**Project administration:** Simanta Roy, Mohammad Azmain Iktidar, Mohammad Delwer Hossain Hawlader.

**Resources:** Shahidur Rahman, Rifat Ara, Mohammad Delwer Hossain Hawlader.

**Supervision:** Simanta Roy, Mohammad Azmain Iktidar, Sreshtha Chowdhury, A. M. Khairul Islam, Shresta Chowdhury, Madhuritu Bhadra Medha, Afia Tasnim, Rifat Ara, Mohammad Delwer Hossain Hawlader.

**Validation:** Simanta Roy, Mohammad Azmain Iktidar, Sreshtha Chowdhury, A. M. Khairul Islam, Auditia Deb, Shresta Chowdhury, Shahidur Rahman, Madhuritu Bhadra Medha, Antara Das Gupta, Afia Tasnim, Mohammad Delwer Hossain Hawlader.

**Visualization:** Rifat Ara, Mohammad Delwer Hossain Hawlader.

**Writing – original draft:** Simanta Roy, Mohammad Azmain Iktidar, Sreshtha Chowdhury, A. M. Khairul Islam, Auditia Deb, Shresta Chowdhury, Shahidur Rahman, Madhuritu Bhadra Medha, Antara Das Gupta, Afia Tasnim, Rifat Ara.

**Writing – review & editing:** Simanta Roy, Rifat Ara, Mohammad Delwer Hossain Hawlader.

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
