## [Decision Letter · Decision Letter 0]

12 Jan 2022

PONE-D-21-40233Prevalence of Dermatological Manifestations due to Face Mask Use during COVID-19 and its' Associated Factors: A Cross-sectional Survey among the General Population of BangladeshPLOS ONE

Dear Dr. Ara,

Thank you for submitting your manuscript to PLOS ONE. After careful consideration, we feel that it has merit but does not fully meet PLOS ONE’s publication criteria as it currently stands. Therefore, we invite you to submit a revised version of the manuscript that addresses the points raised during the review process.

ACADEMIC EDITOR: Title: Please change from: its’ To: itsTitle: Please write the title in a Sentence form. Look for titles of published articles in PLOS ONE.Sampling: Please describe the ‘fixed-step sampling technique’ and report what type of sampling method it is.Results: Please avoid repetition of data both in text and tables such as Table 1 and Text (lines 123-145)..Reporting of P values: Table 2 includes p values reported in in 3, 4 and 5 digits. Please report exact p-values for all values greater than or equal to 0.001. P-values less than 0.001 may be expressed as p < 0.001, or as exponentials in studies of genetic associations. For more details, please refer to statistical reporting for PLOS ONE available at   https://journals.plos.org/plosone/s/submission-guidelines.Please report how was obesity defined and determined.Please report strengths and limitations of the study.

We look forward to receiving your revised manuscript.

Kind regards,

Syed Ghulam Sarwar Shah, M.B.B.S., M.A., M.Sc., Ph.D.

Academic Editor

PLOS ONE

Journal Requirements:

Reviewers' comments:

Reviewer's Responses to Questions

**Comments to the Author**

1. Is the manuscript technically sound, and do the data support the conclusions?

Reviewer #1: Yes

Reviewer #2: Yes

2. Has the statistical analysis been performed appropriately and rigorously? 

Reviewer #1: Yes

Reviewer #2: Yes

3. Have the authors made all data underlying the findings in their manuscript fully available?

Reviewer #1: Yes

Reviewer #2: Yes

4. Is the manuscript presented in an intelligible fashion and written in standard English?

Reviewer #1: Yes

Reviewer #2: Yes

5. Review Comments to the Author

Reviewer #1: The manuscript is technically sound and author has made all data in manuscript fully available. The manuscript is well written and in intelligent manner. However few references are not in Vancouver style, which need to be revised. Reference no 11, 14, 15, 16, 17, 18, 24. The new idea from the study is that the dermatoloical manifestion with face mask are higher and usually this was seen in the cases with wearing of surgical mask and when they are used for longer duration as well as those person who are obese or having any skin problem so for prevention of all these problems there is strong need to take the dermatoloical expert opinion regarding the quality type of face mask in the mass population.

Reviewer #2: The article is a survey study questioning the dermatological findings related to the use of face mask.

The article is written in a generally understandable flowing style. However, some minor errors were detected.

I suggest the following corrections.

- In the sentence of 'Surgical mask users reported more complaints than regular masks, and prolonged use caused more skin symptoms.' on line 35, regular mask can be specified more clearly and other expression can be used instead of regular, which may be better if it includes other masks in parentheses.

- In the survey, it should be clearly stated how the definition of obesity is made. Is it according to the participant's own definition, according to the interviewer, or body mass index? In addition, participants who are obese are not specified in Table 1.

- There are 4 types of masks in Table 1, the other mask is not specified.

- I think in line 152, figure 1 is placed in the wrong place.

- In Table 2, which statistical test is used should be indicated as a note under the table.

- In Table 2, the p value related to others and age is not specified.

- In Table 2, the p value between Average facemask use per week and acne could be <0.001 instead of 0.0005. Again, it would be better if other 5-digit p values were written as 4-digits.

- p<0.001 might be better than p=0.000 in line 172 and table 3.

6. PLOS authors have the option to publish the peer review history of their article (what does this mean?). If published, this will include your full peer review and any attached files.

Reviewer #1: **Yes: **Prof Dr Meharunnissa Khaskheli

Reviewer #2: No

---

## [Author Response · Author response to Decision Letter 0]

23 Jan 2022

We thank the editor and the two reviewers for their valuable comments on our manuscript. We tried our best to response each point raised by the academic editor and reviewers. We hope that we satisfyingly addressed them and the manuscript will be now suited for publication in your journal.

Best regards.

---

## [Decision Letter · Decision Letter 1]

1 Apr 2022

PONE-D-21-40233R1Prevalence of Dermatological Manifestations due to Face Mask Use and its Associated Factors during COVID-19 among the General Population of Bangladesh: A Nationwide Cross-sectional SurveyPLOS ONE

Dear Dr. Ara,

Thank you for submitting your manuscript to PLOS ONE. After careful consideration, we feel that it has merit but does not fully meet PLOS ONE’s publication criteria as it currently stands. Therefore, we invite you to submit a revised version of the manuscript that addresses the points raised during the review process.

 Please submit your revised manuscript by May 16 2022 11:59PM. If you will need more time than this to complete your revisions, please reply to this message or contact the journal office at plosone@plos.org. Please include the following items when submitting your revised manuscript:A rebuttal letter that responds to each point raised by the academic editor and reviewer(s). You should upload this letter as a separate file labeled 'Response to Reviewers'.A marked-up copy of your manuscript that highlights changes made to the original version. You should upload this as a separate file labeled 'Revised Manuscript with Track Changes'.An unmarked version of your revised paper without tracked changes. You should upload this as a separate file labeled 'Manuscript'.If applicable, we recommend that you deposit your laboratory protocols in protocols.io to enhance the reproducibility of your results. Protocols.io assigns your protocol its own identifier (DOI) so that it can be cited independently in the future. For instructions see: https://journals.plos.org/plosone/s/submission-guidelines#loc-laboratory-protocols. Additionally, PLOS ONE offers an option for publishing peer-reviewed Lab Protocol articles, which describe protocols hosted on protocols.io. Read more information on sharing protocols at https://plos.org/protocols?utm_medium=editorial-email&utm_source=authorletters&utm_campaign=protocols.

We look forward to receiving your revised manuscript.

Kind regards,

Syed Ghulam Sarwar Shah, M.B.B.S., M.A., M.Sc., Ph.D.

Academic Editor

PLOS ONE

Journal Requirements:

Additional Editor Comments (if provided):

Please address the issues raised by the reviewer(s).

Reviewers' comments:

Reviewer's Responses to Questions

**Comments to the Author**

1. If the authors have adequately addressed your comments raised in a previous round of review and you feel that this manuscript is now acceptable for publication, you may indicate that here to bypass the “Comments to the Author” section, enter your conflict of interest statement in the “Confidential to Editor” section, and submit your "Accept" recommendation.

Reviewer #1: All comments have been addressed

Reviewer #2: All comments have been addressed

Reviewer #3: All comments have been addressed

2. Is the manuscript technically sound, and do the data support the conclusions?

Reviewer #1: Yes

Reviewer #2: Yes

Reviewer #3: Yes

3. Has the statistical analysis been performed appropriately and rigorously? 

Reviewer #1: Yes

Reviewer #2: Yes

Reviewer #3: Yes

4. Have the authors made all data underlying the findings in their manuscript fully available?

Reviewer #1: Yes

Reviewer #2: Yes

Reviewer #3: Yes

5. Is the manuscript presented in an intelligible fashion and written in standard English?

Reviewer #1: Yes

Reviewer #2: Yes

Reviewer #3: Yes

6. Review Comments to the Author

Reviewer #1: Manuscript is fulfilling the criteria of Plosone , accepted to for publication. This is good effort best research work by the author for writing the manuscript and fulfilling the criteria of manuscript for the joural. So this acceptableb for the publication.

Reviewer #2: (No Response)

Reviewer #3: Recommendations reviewer

I think that there is an important and relevant story in this study and I recommend your publication. The prevalence of dermatological manifestations due to face mask use during covid-19 is a very relevant theme. However minor revision is necessary.

Method

A structured questionnaire was utilized to evaluate fot Facemask-related dermatological. This questionnaire was submittig to evaluate for specialists before to study? Or carried out pilot test?

Discussion

In the line 237: From the opinions about the preference of face mask types, cotton cloth masks were more 238 preferable and comfortable among our study participants. Rather, dermatological problems were 239 profound among the surgical facemask users. This outcome is noteworthy, as this predilection 240 significantly can help in decreasing the demand for surgical masks among general people and 241 should be reserved for the healthcare providers during the COVID-19 pandemic. A A prospective 242 survey was conducted in Thailand, where the effects of facemask use have been compared among 243 the healthcare providers and non-healthcare general citizens (24). This survey showed some 244 similar outcomes along with a higher risk of skin problems among the healthcare workers. This is 245 very obvious, as health care workers (HCW) use facemask more frequently and for a longer 246 duration than the non-HCWs.

The comparison with health professionals and the time of use of masks with population data does not seem to be fair, as several factors must be taken into account, as well as the time of use, which may be longer among some individuals in the general population who works for long periods with masks depending on their work activity, not just health professionals.

7. PLOS authors have the option to publish the peer review history of their article (what does this mean?). If published, this will include your full peer review and any attached files.

Reviewer #1: **Yes: **Meharunnissa Khaskheli

Reviewer #2: No

Reviewer #3: **Yes: **Fernanda Ávila

---

## [Author Response · Author response to Decision Letter 1]

5 Apr 2022

Respected Reviewers and Editor,

Thank you for your valuable comments. We hope that we satisfyingly addressed them and that the manuscript will be now suited for publication in your journal.

Best regards.

---

## [Decision Letter · Decision Letter 2]

1 Jun 2022

Prevalence of Dermatological Manifestations due to Face Mask Use and its Associated Factors during COVID-19 among the General Population of Bangladesh: A Nationwide Cross-sectional Survey

PONE-D-21-40233R2

Dear Dr. Ara,

We’re pleased to inform you that your manuscript has been judged scientifically suitable for publication and will be formally accepted for publication once it meets all outstanding technical requirements.

Kind regards,

Syed Ghulam Sarwar Shah, M.B.B.S., M.A., M.Sc., Ph.D.

Academic Editor

PLOS ONE

Additional Editor Comments (optional):

Many thanks for submitting your revised manuscript and successfully addressing all issues raised by the academic editor and reviewers.

Reviewers' comments:

Reviewer's Responses to Questions

**Comments to the Author**

1. If the authors have adequately addressed your comments raised in a previous round of review and you feel that this manuscript is now acceptable for publication, you may indicate that here to bypass the “Comments to the Author” section, enter your conflict of interest statement in the “Confidential to Editor” section, and submit your "Accept" recommendation.

Reviewer #3: All comments have been addressed

2. Is the manuscript technically sound, and do the data support the conclusions?

Reviewer #3: Yes

3. Has the statistical analysis been performed appropriately and rigorously? 

Reviewer #3: Yes

4. Have the authors made all data underlying the findings in their manuscript fully available?

Reviewer #3: Yes

5. Is the manuscript presented in an intelligible fashion and written in standard English?

Reviewer #3: Yes

6. Review Comments to the Author

Reviewer #3: All recommendations have been carried out. The manuscript is relevant and important. I consider accepted for the publication.

7. PLOS authors have the option to publish the peer review history of their article (what does this mean?). If published, this will include your full peer review and any attached files.

Reviewer #3: **Yes: **Fernanda Ávila

---

## [Editor Report · Acceptance letter]

3 Jun 2022

PONE-D-21-40233R2 

Prevalence of Dermatological Manifestations due to Face Mask Use and its Associated Factors during COVID-19 among the General Population of Bangladesh: A Nationwide Cross-sectional Survey  

Dear Dr. Ara:

I'm pleased to inform you that your manuscript has been deemed suitable for publication in PLOS ONE. Congratulations! Your manuscript is now with our production department. 

Kind regards, 

on behalf of

Dr. Syed Ghulam Sarwar Shah 

Academic Editor

PLOS ONE